# BAZ1B the Protean Protein

**DOI:** 10.3390/genes12101541

**Published:** 2021-09-28

**Authors:** Shahin Behrouz Sharif, Nina Zamani, Brian P. Chadwick

**Affiliations:** 1Department of Biological Science, Florida State University, Tallahassee, FL 32306, USA; ssharif@bio.fsu.edu; 2Institute of Molecular Biophysics, Florida State University, Tallahassee, FL 32306, USA; nzamani@fsu.edu

**Keywords:** WSTF, BAZ1A, BAZ: WICH, B-WICH, hSNF2H, ISWI, chromatin remodeling

## Abstract

The bromodomain adjacent to the zinc finger domain 1B (BAZ1B) or Williams syndrome transcription factor (WSTF) are just two of the names referring the same protein that is encoded by the *WBSCR9* gene and is among the 26–28 genes that are lost from one copy of 7q11.23 in Williams syndrome (WS: OMIM 194050). Patients afflicted by this contiguous gene deletion disorder present with a range of symptoms including cardiovascular complications, developmental defects as well as a characteristic cognitive and behavioral profile. Studies in patients with atypical deletions and mouse models support BAZ1B hemizygosity as a contributing factor to some of the phenotypes. Focused analysis on BAZ1B has revealed this to be a versatile nuclear protein with a central role in chromatin remodeling through two distinct complexes as well as being involved in the replication and repair of DNA, transcriptional processes involving RNA Polymerases I, II, and III as well as possessing kinase activity. Here, we provide a comprehensive review to summarize the many aspects of BAZ1B function including its recent link to cancer.

## 1. Introduction

The Williams syndrome transcription factor (WSTF) encoded by the WBSCR9 gene was first reported in 1998 by two independent studies [1,2]. It was described as a single copy gene spanning 80 kb with a 4449 bp open reading frame encoding a protein of 1483 amino acids. The gene structure is composed of two isoforms with either 19 or 20 exons, with the 20-exon isoform differing by splicing to an extra 3′ non-coding exon that increases the size of the 3′ untranslated region (Figure 1). Exons range from 99 bp for exon 11 to 1702 bp for exon 7 and are flanked by the standard GT-AG splice donor and acceptor sequences. The gene is located at 7q11.23 and is oriented 5′ telomeric and 3′ centromeric. Through the use of florescence in situ hybridization, WBSCR9 was physically mapped to the Williams syndrome (WS) deletion region [1]. Williams Syndrome is a contiguous gene deletion syndrome (OMIM 194050) that causes a complex developmental disorder that presents with multi-systemic defects. Intellectual disability, dysmorphic facial features, a unique cognitive profile along with congenital cardiovascular complications, infantile hypercalcemia, and growth deficiency are common characteristics of WS patients [3,4]. The syndrome results from a heterozygous deletion of >1 Mb (up to 1.83 Mb) mapping to 7q11.23 (Figure 1), which is prone to non-allelic homologous recombination and encompasses roughly 28 genes [5,6]. Duplication of the same interval in 7q11.23 Duplication Syndrome (OMIM 609757) results in individuals being triploid for most of the same genes and is characterized by numerous phenotypes including cardiac abnormalities, craniofacial developmental defects, anxiety, and autistic spectrum disorder [7].

WSTF is universally expressed, with the highest transcript levels being detected in adult brain, heart, ovary, placenta, and skeletal muscle tissues (Figure 2).

It also forms physical associations with several other important proteins. Figure 3 illustrates the network of the functional and physical associations between BAZ1B and the first shell of interactors as well as their observed co-expression in *Homo sapiens* (Figure 3).

The WSTF protein has a molecular mass of 171 kDa and was speculated by Lu et al. to function as a transcription factor based upon the existence of conserved motifs in the protein sequence (Figure 4). In addition to the conserved motifs, several phosphorylation, amidation, 13 N-myristylation, and glycosylation sites as well as a specific acidic amino acid-enriched region (amino acids 1260–1274) were annotated [1]. Additionally, there is a polyglutamate stretch (13 residues) and several nuclear localization signals (three types) that support the protein to function in the nucleus. Northern blot hybridization analysis has also revealed an alternatively spliced smaller 4.5 kb message that excludes the 1.7-kb exon 7 with comparable expression levels to the larger transcript in the liver, lung, and kidney. However, the larger transcript is predominantly expressed in the brain and placenta [2].

### 1.1. Bromodomain

Within the C-terminus of WSTF is a bromodomain (BrD) that spans 71 amino acids from residues 1356 to 1426, contains a consensus sequence and helix-turn-helix structure, and is required for protein–protein interactions [1,2] (Figure 4).

The BrD is an evolutionary conserved structural domain in proteins that bind acetylated lysine (Kac) residues [11,12,13]. This interaction is mediated by a highly conserved Asparagine residue within the BrD. For instance, it has been demonstrated through the crystal structure that the formation of a hydrogen bond between the amide nitrogen of Asn407 (the conserved asparagine in the BrD of Saccharomyces cerevisiae Gen5p bromodomain complex) and the oxygen of the acetyl carbonyl group is a key event in the BrD-mediated protein–protein interaction between Gen5p and histone H4 acetylated at lysine 16 [14]. BrD proteins mediate protein–protein interactions in DNA recombination, replication, and repair; in histone modifications and chromatin remodeling; as well as in the recruitment of transcription factors that affect both transcription initiation and elongation processes [15]. The inhibition of BrD and extra-terminal domain (BET) family proteins prevents proliferation in carcinoma cells [16] and blocks the expression of inflammatory genes [17].

**Figure 4 genes-12-01541-f004:**
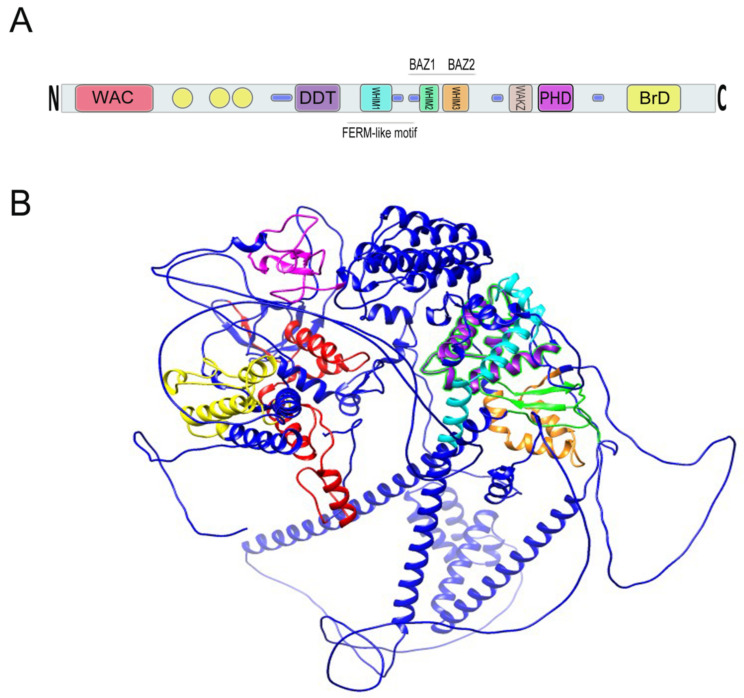
(**A**) Relative location of BAZ1B domains. Yellow circles are low complexity regions. Blue cylinders are coiled coil regions. The first N-terminally located domain is present in Acf1-related proteins in a variety of organisms. The WAC (WSTF/Acf1/cbp146) domain is within the DNA binding region of the Acf1 and is believed to be involved in DNA binding and is associated with the PHD finger and Brd domains in other proteins [18,19]. It spans residues 21–120 on BAZ1B. DDT stands for DNA-binding homobox-containing proteins and Different Transcription and chromatin remodeling factors. Such proteins share this domain, which spans almost 60 residues and is associated exclusively with nuclear domains including PHD finger, Brd, and DNA-binding homeodomain [20]. It spans residues 604–668 on BAZ1B. Residues 724–773 on BAZ1B locate WHIM1 conserved the *α* helical motif that along with the WHIM2, WHIM3, and DDT domain comprise an *α* helical module that is reported to interact with linker DNA and the SLIDE domain of ISWI proteins [21]. WHIM2 domain spans residues 899 to 936 on BAZ1B and contains the D-TOX E motif that is also known as the Williams–Beuren Syndrome DDT (WSD) motif. It is conserved from yeast to animals [22]. PHD zinc finger (spanning 1184 to 1234) and bromodomain (1356–1426) are the two C-terminally located domains. Refer to the main text for more information about these domains. (**B**) A 3D representation of BAZ1B based on its PDB file (PDB ID: Q9UIG0) predicted by the AlphaFold protein structure database [23]. Domains are highlighted with different colors as follows: BrD: yellow; PHD: magenta; WHIM1: cyan; WHIM2: green; WHIM3: orange; DDT: purple; WAC: red.

### 1.2. Plant Homeodomain Finger

The second conserved motif directly adjacent to the BrD is a cysteine/histidine-rich plant homeodomain type (PHD-type) zinc finger motif (Figure 4) that encompasses 51 residues (1184–1234) and that contains the conserved characteristics of a typical PHD finger [1,2].

PHD fingers are structurally conserved zinc-finger-like motifs with a unique Cys4-His-Cys3 pattern that is distinct from that of similar sized RING finger and LIM domain zinc fingers [24] (Figure 5). These versatile epigenome readers interact with the first six N-terminal residues of histone H3. Several studies have implicated intriguing and complicated functions of PHD fingers in reading histone codes such as histone H3 tri-methylated at lysine 4 (H3K4me3), unmethylated (H3K4me0), arginine 2 of histone H3 (H3R2), and histone H3 acetylated at lysine 14 (H3K14ac). A single PHD finger can even recognize a combination of histone post-translation modifications by using multiple binding sites within the motif. For example, double tandem PHD Finger protein 3b (DPF3b), reads the combination of H3K14ac, H3K4me0 and H3R2 [25]. In addition to the BrD, PHD fingers are highlighted as epigenetic modulators of transcription via the recruitment of chromatin remodeling and transcription factors as well as basal transcription machinery [12,24,25].

### 1.3. WAKZ and WAC Motifs

In addition to the BrD and PHD finger, there is a WAKZ (WSTF/Acf1/KIAA0314/ZK783.4) motif located immediately proximal to the PHD finger as well as a WAC (WSTF/Acf1/cbp146) motif at the amino terminus of WSTF that spans 107 residues (20–126) (Figure 4). These two motifs were first described by Ito et al. in the ATP-dependent chromatin assembly factor 1 (Acf1) protein of *Drosophila melanogaster*. They described Acf1 as a novel protein containing two PHD fingers, a bromodomain, a WAKZ, and a WAC domain [18].

WAKZ is a common motif between members of the highly conserved bromodomain adjacent zinc finger, the BAZ family in humans, as well as their homolog in *Drosophila* (Acf1) and *Caenorhabditis* (ZK783.4) [13]. Chromatin assembly is slightly defective when the WAKZ, PHD finger, and BrD domains are mutated in Acf1; however, the binding of the ACF complex (Acf1/ISWI) to DNA is not impaired upon the mutation of either of these regions [19]. In addition to the WAKZ motif, the WAC motif located in the N-terminal is also a shared feature between members of the BAZ family and Acf1 [13]. Concordantly, Poot et al. identified these proteins as the WAL (WSTF/Acf1 Like) family. WAC motifs are required for DNA binding and may be sufficient for WSTF to target heterochromatin in pericentric regions [19]. The WAC domain in conjunction with the adjacent C-Motif (amino acids 206–345) have also been reported to function as a tyrosine kinase [27] (See Section 3.1 below).

### 1.4. WSTF or BAZ1B?

The BAZ gene family was first described as a novel BrD family by Jones et al. that has two subfamilies when considering their conserved residues. The BAZ1 subfamily involves BAZ1A (14q12-q13) and BAZ1B (7q11-q21), whereas the BAZ2 subfamily is comprised of BAZ2A (12q24.2-qter) and BAZ2B (2q23-q24). Based on a high degree of sequence conservation, the BAZ1 subfamily is closely related to Acf1, and the BAZ2 subfamily is related to the baz-2 (ZK783.4) of *Caenorhabditis elegans* and are considered orthologues [13]. BAZ1B is more related to human Acf1 (hACF1 or BAZ1A) than BAZ2A and BAZ2B are. The common WAC motif at the N-terminus of the proteins is also another distinct feature of the BAZ1 subfamily compared to more distant members of the WAL family [28]. Northern analysis has confirmed the expression of approximately 7.5 kb and 9.5 kb transcripts of BAZ1 and BAZ2 genes, respectively, in a range of tissues with markedly variable expression levels. For instance, BAZ1A is highly expressed in the testis and BAZ2A is moderately expressed in most analyzed tissues. This might confer tissue-specific transcriptional regulatory functions. All BAZ members have seven highly conserved motifs, out of which six are shared among all BAZ members. The WAC domain is present in BAZ1/Acf1 proteins but not in BAZ2/ZK783.4; on the other hand, a ZB2 domain is shared between BAZ2/ZK783.4 but not BAZ1/Acf1 proteins. BAZ1B, the second member of the BAZ1 subfamily reported by Jones et al., is WSTF and, therefore, from this point forward, WSTF shall be referred to as BAZ1B for the remainder of this review.

### 1.5. BAZ or WHIM Motifs?

In addition to the abovementioned BrD, PHD finger, WAKZ, and WAC motifs, Jones et al. also identified BAZ1 and BAZ2 (Figure 4) as two other conserved motifs between all BAZ members as well as the Acf1 and Zk783.4 proteins [13]. More than a decade later, Aravind and Iyer also reported a novel N-terminal domain in the human ASLX protein and named it HARE-helix-turn-helix (HTH) domain after the proteins HB1, ASLX, and Restriction Endonuclease, the proteins in which it was detected. The HARE-HTH leucine-rich helical domain is not only present in several eukaryotic chromatin proteins but also in many prokaryotic key factors such as restriction endonucleases and RNA polymerase complexes. There are three helices within the HTH unit that form a distinct type of “wing helix-turn-helix”. HARE-HTH displays a specific conservation pattern, and considering its highly conserved structure in various essential proteins in human, fish, plants, chlorophyte algae, and red algae, it has been presented as indispensable for multidomain proteins that are involved in chromatin-related functions. It was also postulated that some HARE-HTH proteins might identify specific DNA modifications such as 5-methyl-cytosine [21]. HARE-HTH is associated with a DDT motif (explained below in 1–6) and either of the homeo- or WAC motifs.

In the BAZ family of proteins, HARE-HTH, DDT, and WAC motifs are present and are separated by low-complexity regions. This pattern is also true of other factors such as plant HB1 and chlorophyte HDZ1, where HARE-HTH is associated with the DDT and homeodomain instead of WAC. There are three HARE-HTH motifs on BAZ1B (Figure 4), the second and third of which overlap with the previously described conserved BAZ1 and BAZ2 motifs. These motifs were named WHIM (WSTF, HB1, Itc1p, MBD9) motifs 1, 2, and 3. Acf1 interaction with SNF2 is mediated by a region containing a DDT and three WHIM motifs, and it is postulated that ISWI binding is a common feature of WHIM-containing proteins. The BAZ1B WHIM1, 2, and 3 motifs span residues 724–773, 899–936, and 991–1032, respectively. WHIM is significantly associated with the DDT motif, and together, they form a binding pocket for the SLIDE domain of ISWI (Figure 6). Highly conserved residues in WHIM1 work cooperatively with DDT and WHIM2 and mediate interaction with ISWI. WHIM motifs coupled with DDT domain are postulated as “protein rulers” that can regulate nucleosome spacing [21].

### 1.6. DDT and LXXLL Motifs

The DDT domain, which is found in DNA-binding homeobox-containing proteins and different transcription and chromatin remodeling factors was first described by Doerks et al. through homology-based sequence analysis [20]. It consists of ~60 residues with conserved N-terminal phenylalanines, several aromatic and charged residues, and C-terminal leucines. It contains three α helices and exclusively associates with common domains found in nuclear proteins. In addition to bromodomain PHD finger transcription factors (BPTF) and putative DNA-binding proteins, DDT has also been identified in the BAZ family.

A leucine-rich helix domain was also reported by Jones et al. and was called the LH domain, which is a conserved region and contains a Leu-Xaa-Xaa-Leu-Leu (LXXLL) motif in an almost identical position between the WAC and BAZ1 domains among all BAZ members. BAZ1B displays two tandemly arranged LXXLL motifs where BAZ1BL, which is a structural variant of BAZ1B because it has 12 additional nucleotides in exon 7, harbors an additional LXXLL motif next to the original two [13]. The LXXLL motif was first reported in 1997 as a short, conserved motif that is capable of and sufficient to bind transcriptionally active nuclear receptors [29]. These nuclear receptor box motifs have been implicated in several bromodomain proteins and are suggested to be responsible for binding to nuclear receptors for retinoic acid, estrogen, progesterone, and vitamin D3 [30]. For instance, there are three LXXLL motifs within the core interaction domain of human SRC-1a that mediate its binding to estrogen receptors (ER) [29].

### 1.7. Other Motifs and Conserved Regions

A FERM domain has also been identified toward the middle of BAZ1B between the DDT and BAZ1 domains [31] (Figure 4). This domain has been assigned several different names, including the amino-terminal domain, membrane-cytoskeletal-linking domain, erzin-like domain of the band 4.1 superfamily, the ERM like domain, and the conserved N-terminal domain; however, FERM, which stands for *4.1 protein erzin radixin meosin,* has subsequently been proposed as a consensus naming format [32]. It contains multiple phosphorylation and N-linked glycosylation sites and is mainly composed of β-sheet structures in addition to a few α-helices. This hydrophobic cysteine-rich domain is involved in protein-mediated cytoskeleton attachment to the plasma membrane and in maintaining cell integrity and mobility [33]. The presence of this domain in a nuclear protein is somewhat unexpected, and what function, if any, it performs in BAZ1B remains unclear.

Two PEST sequences were identified in BAZ1B [2] that are common among proteins with a short intracellular half-life. Many PEST sequences are conditional proteolytic signals for rapid degradation that can be activated in several ways. These sequences are enriched in proline (P), glutamic acid (E), serine (S), and threonine (T) residues; are uninterrupted but confined by positively charged amino acids (lysine, arginine, or histidine); and are found in various functionally important proteins such as oncogenes (e.g., adenovirus early region 1A (E1A), c-fos, p53), transcription factors (e.g., c-myc, v-myb), key metabolic enzymes (e.g., ornithine decarboxylase (ODC)), cyclins, and protein phosphatases [34,35]. A number of PEST domains have been implicated as anchor sites for the E3 ubiquitin ligases involved in unbiquitin-mediated protein degradation [36]. Considering their presence in long-lived proteins too, alternative functions are also suggested for PEST domains including intracellular sorting, binding of the SUMO-conjugating enzyme Ubc9, and modulation of the inward-rectifier potassium channel *IK1* [37].

## 2. BAZ1B and Chromatin Remodeling

Nucleosome assembly packages and organizes the genome; however, as a result, access to many regulatory DNA elements is also blocked. For the sake of diverse chromatin-related processes within the nucleus, nucleosome composition has to be tailored in order to grant dynamic access to the packaged DNA wrapped around the histone octamers. This process is conducted by chromatin remodelers.

### 2.1. WICH: A Chromatin Remodeling Complex for Replication

Two years after the discovery of BAZ1B, Bozhenok described the WSTF-ISWI chromatin remodeling (WICH) complex after purifying it from *Xenopus* egg extracts [31]. BAZ1B and human sucrose nonfermenting protein 2 homolog (SNF2H, encoded from the *SMARCA5* gene) are the subunits of the WICH complex. Each of the mammalian chromatin remodeling complexes contains one of the two members of the SNF2 family, SNF2L or SNF2H [38]. SNF2 is classified as the ATPase subunit of the human chromatin remodeling complexes and is an orthologue of the *D. melanogaster* imitation switch (ISWI) family of ATP-dependent chromatin remodelers [39]. ISWI remodelers are highly conserved among eukaryotes and orchestrate nucleosome assembly and spacing in addition to higher order chromatin organization. All ISWI proteins have ATPase domains, which consist of DExx and HELICc regions as well as DNA binding activity through their HAND/SANT/SLIDE domain in their C-terminus [40]. BAZ1A (hACF1) is also another well-known partner of SNF2H that together form the ACF complex, which regulates inter-nucleosomal spacing in chromatin [18]. Figure 6 illustrates the SANT domain of SNF2H bound to DNA where BAZ1B is approaching it via the DDT, WHIM1, WHIM2, and WHIM3 domains together as a binding pocket for SNF2H (Figure 6).

The WICH complex is conserved in vertebrate phyla, and in mouse cells, it was demonstrated by confocal microscopy that BAZ1B is nuclear and dynamically associates with pericentromeric heterochromatin while the underlying DNA is in the process of replication. In synchronization experiments, its enrichment at these regions begins before G1/S release, where it is excluded after one hour and is redistributed more evenly in a granular pattern throughout the nucleoplasm before becoming markedly enriched again at these foci after the replication of pericentromeric heterochromatin (5 h into S phase). Pericentromeric heterochromatin is crucial for chromosome stability [41] and is characterized by histone hypoacetylation along with DNA hypermethylation. BAZ1B has been speculated to facilitate heterochromatin replication or its assembly post-replication [31].

In BAZ1B depleted cells, levels of H3K9me3 and H3K27me2 are increased 1.5-fold in addition to the slightly increased acetylation of histone H4 at lysines 8 and 12. Although BAZ1B is not involved in the repression of heterochromatin protein 1 (HP1) expression, the depletion of BAZ1B culminates with HP1α and HP1β (heterochromatin markers) being increased by 1.5–2 and 2.5–3 folds, respectively. However, HP1γ remains unaffected [42]. This increase in the levels of HP1 is associated with BAZ1B-mediated chromatin remodeling at replication sites. Newly replicated DNA is less readily digested in cells with reduced levels of BAZ1B. The depletion of BAZ1B has been demonstrated to be the most impactful on chromatin assembly within the first ten minutes immediately following DNA replication. BAZ1B depletion also results in the appearance of smaller nuclei compared to wild type cells (Poot et al., 2004), which is similar to the phenotype observed in cells with ectopic expression of the myeloid and erythroid nuclear termination stage specific (MENT) protein [43].

Consistently, a unique nuclear phenotype has also been observed in human cells lacking BAZ1B. A substantial proportion of cells showed the acquisition of large 4′,6-diamidino-2-phenylindole (DAPI)-dense staining masses throughout the nucleus [44], many of which were comparable in size to the Barr body [45]. These DAPI-dense masses display the dynamic enrichment of heterochromatin features such as elevated levels of H3K9me3, HP1, and the heterochromatin protein SMCHD1 (structural maintenance of chromosomes flexible hinge domain containing 1) and are present for extensive periods during the cell cycle. Nevertheless, these blocks are transient and resolve before proceeding with the cell cycle. In addition to transient heterochromatin block formation, stable alterations in global gene expression have been observed, with both up and down regulation in the BAZ1B knockouts [44].

We have previously reported the transient association of BAZ1B as part of the WICH complex with chromatin of the inactive X chromosome (Xi) immediately before the detection of elevated levels of histone variant H2AX phosphorylation at serine 139 (γH2A.X) and the breast cancer 1 (BRCA1) protein [46]. However, the loss of BAZ1B does not appear to impact the maintenance of the Xi [44], and therefore, the enrichment at this site may simply reflect the standard role of WICH at replication foci that in the case of the Xi is unusually large due to the unique replication profile of the Xi, whereupon numerous multi-megabase stretches of the chromosome replicate simultaneously alongside other considerably smaller heterochromatic regions of the autosomes during mid-late S-phase [47].

BAZ1B interacts with the proliferating cell nuclear antigen (PCNA) through its four “PCNA-binding motifs” (residues 664–671, 1026–1033, 1101–1108, and 1434–1441) and is recruited to replication foci followed by BAZ1B-mediated SNF2H localization. This is thought to maintain an open chromatin structure [42]. PCNA is an essential factor in the processivity of DNA polymerase and regulates various steps associated with replication [48].

Throughout the S-phase, from the very early until the very late stages, BAZ1B co-localizes with bromodeoxyuridine incorporation sites at replication foci and remains associated with chromatin for an additional 30–60 min after replication. Pull-down experiments have demonstrated that the WICH complex is retained at replication sites and co-precipitates with PCNA [42]. By improving nucleosome mobilization, WICH is thought to provide chromatin accessibility and an opportunity for factors such as the transcription regulators and chromatin modifying enzymes involved in epigenetic maintenance to bind newly replicated DNA and to re-establish chromatin marks. This is required in order to maintain open chromatin and to prevent inappropriate new heterochromatin formation immediately after replication. Although they are not necessarily bound to the nascent DNA, PCNA and BAZ1B localize close to its vicinity and are involved in the maturation of nascent chromatin into higher-order fibers [49]. The PCNA-mediated recruitment of WICH to DNA highlights its global interaction with nascent chromatin irrespective of position and sequence. PCNA might also act as a processivity factor for WICH. It was later shown that WICH can also localize in order to replicate heterochromatin in a PCNA-independent manner [46].

In Hela cell nuclear extracts, the WICH complex binds to the chromatin licensing and DNA replication factor Cdc10-dependent transcript1 (CDT1). Moreover, SNF2H has also been demonstrated to co-immunoprecipitate with ectopically expressed CDT1 [50]. CDT1 has a crucial role in regulating DNA replication and is required to be degraded in S phase to limit DNA replication to just one round. DNA re-replication and chromosomal damage can result from CDT1 persistence and, similar to BAZ1B, CDT1 directly interacts with PCNA via its “PCNA interaction motifs” [51]; however, the association between WICH and CDT1 is not through their association with PCNA but is instead a direct interaction through SNF2H. BAZ1B localizes to replication foci in a cell cycle-dependent manner and results in excess MCM helicase loading, which is mediated by interaction with CDT1. However, this interaction is not required for minimal MCM loading [52]. SNF2H is involved in the stimulation of DNA replication and is implicated by its function in the effective initiation of SV40 DNA replication [53].

### 2.2. B-WICH: A Transcription Regulator Complex

Several years later, it was reported that BAZ1B is involved in RNA Pol-I and Pol-III transcription via a unique variant of the complex called B-WICH [54,55]. B-WICH is a distinct 3-MDa assembly that is detected during active transcription and includes six additional proteins that interact with the core components of the WICH complex (BAZ1B/SNF2H).

One of the B-WICH subunits is Cockayne syndrome protein B (CSB), which has decisive roles in transcription coupled repair (TCR), base excision repair (BER), RNA Pol-I [56] and RNA Pol-II [57] transcription, and neurogenesis [58] (Ciaffardini et al., 2014). Nuclear RNA helicase II (DDX21) is another subunit that plays an essential role in ribosomal RNA (rRNA) transcription and processing and binds various RNA species including rRNA, snoRNA, 7SK, and mRNA [59,60] (Henning et al., 2003, Calo et al., 2015). Splicing factor 3B subunit 1 (SF3b1), which is essential for the branch site selection contributing to pre-mRNA splicing [61], and Myb-binding protein 1a (MYBBP1a) are other members. MYBBP1a was first described in 1998 as a homologue of murine p160 and a plausible transcription factor [62]. It has also been suggested to couple ribosome biogenesis to the Myb-dependent transcription that regulates the cell cycle [63]. Protein DEK, which is an abundant nucleic acid-associated nuclear phosphoprotein, is associated with various human diseases when mutated [64] and can act as proto-oncogene in acute myeloid leukemia [65]. It has also been also identified as a senescence inhibitor and transcriptional target of HPV E7 proteins [66].

Nuclear myosin 1 (NM1 or MYO1C) is an important member of the B-WICH complex. It is a splice isoform of myosin 1b that has an additional 12 amino acids preceding the first methionine of myosin 1b and was first identified to colocalize with RNA Pol-II within the nucleus [67]. NM1 broadly binds the mammalian genome and is required to stabilize the B-WICH complex, which is essential for the SNF2H-mediated correct repositioning of nucleosomes. NM1 also predominantly associates with the plasma membrane and controls its tension [68,69]. The interaction between BAZ1B, SNF2H, and NM1 is thought to be a direct protein–protein interaction since RNase-A treatment can disrupt the association of MYBBP1a, SF3b1, and RNA helicase II with B-WICH but not NM1. The association of MYBBP1a and SF3b1 with the assembly has also been indicated to depend on transcription [54]. NM1 dynamically targets SNF2H and stabilizes B-WICH on the promoter and transcribed regions of ribosomal DNA (rDNA) in a BAZ1B-dependent manner. Once it occupies rDNA, B-WICH recruits histone acetyl transferases (HAT) and maintains the required levels of histone H3 lysine 9 acetylation (H3K9-Ac) in cells close to mitosis exit as well as early G1 cells.

The B-WICH complex co-localizes with the RNA Pol-I complex via actin, and the interaction of actin with NM1 facilitates polymerase movement on permissive chromatin and improves 45S pre-rRNA expression. NM1 interaction with SNF2H and actin is mediated through overlapping sites, and their interaction can be mutually exclusive. B-WICH acts specifically on chromatin since RNAi-mediated BAZ1B knock-down impairs Pol-I function on chromatin but not on naked DNA. BAZ1B phosphorylation leads to B-WICH disassembly during temporarily stalled transcription at the onset of cell division [55,70]. The nuclear mitotic apparatus (NuMA) protein, an important protein in major cellular events such as gene transcription, also co-localizes with B-WICH, RNA Pol-I, and the ribosomal proteins RPL26 and RPL24 and participates in rDNA transcription. The interaction of B-WICH, RNA Pol-I, and other factors might be orchestrated by NuMA, which facilitates the initiation of rDNA transcription [71].

The B-WICH complex not only remodels chromatin structure by actively preventing its compaction but is also involved in facilitating the recruitment of several HATs to rDNA promoters (Figure 7). Such acetylation is facilitated by a chromatin remodeling event on ~200 bp on the rDNA promoter where core promoter elements and upstream control elements are present. In BAZ1B knock down cells, B-WICH occupancy and HAT recruitment to the promoter as well as global histone acetylation levels, specifically H3K9-Ac, are decreased. In serum-starved cells where rRNA transcription is reduced, B-WICH’s association with the rDNA promoter and BAZ1B expression are markedly decreased [72]. The core proteins of B-WICH are also involved in RNA Pol III-transcription activity, including 5S rRNA and 7SL RNA expression but not tRNA genes or other small nuclear RNAs. The shRNA-mediated knock down of BAZ1B results in reduced expression levels of 5S rRNA and 7SL RNA. B-WICH remodels chromatin structure and maintains open chromatin in regions neighboring 5S rRNA and 7SL RNA genes [54].

B-WICH activity is also required for the binding of RNA Pol-II transcription factors such as c-Myc. c-Myc binds to its target sites, which are remodeled and exposed by B-WICH, and activates transcription by inducing histone acetylation [73]. Outside of rDNA, NM1 is also associated with numerous sites in the coding and non-coding regions of the genome, and its distribution correlates with RNA Pol-II occupancy as well as Pol-II related active transcription marks. The recruitment of the HAT p300/CBP-associated factor complex (PCAF) and the histone methyltransferase (HMT) complex Set1/Ash2 to class II promoters are mediated by NM1 following B-WICH assembly and are required to maintain H3K9Ac and H3K4me3 [69].

Cytochrome P450 family 21 subfamily A member 2 (CYP21A2) is a steroid 21- hydroxylase that is expressed in the adrenal cortex and is involved in estrogen and androgen biogenesis. It has been reported that the biologically active form of vitamin D, 1α,25-dihydroxyvitamin D3, decreases CYP21A2 expression and enzymatic activity [74]. Investigating this mechanism, the effects of vitamin D on the CYP21A2 gene promoter has been reported to be mediated by the interaction of the vitamin D receptor (VDR), VDR interacting repressor (VDIR), and BAZ1B. This complex interacts with vitamin D response elements (VDRE) in the promoter region and, in the absence of 1α,25-Dihydroxyvitamin D3, the process involves the recruitment of the BAZ1B/VDR/VDIR complex in addition to histone deacetylation and DNA methylation events. BAZ1B dissociates upon the addition of vitamin D, leading to decreased CYP21A2 expression. Therefore, protein levels of BAZ1B and VDIR can modulate vitamin D-mediated CYP21A2 expression regulation [74].

## 3. BAZ1B and DNA Repair

WICH and ACF both have been demonstrated to localize to UV-C induced DNA damage sites. Their action in maintaining open chromatin facilitates CSB binding and promotes lesion-mediated stalled RNA Pol-II transcription. SNF2H promotes CSB binding to active transcription-coupled repair (TC-NER) complexes and modulates the repair efficiency. It is thus conceivable that ACF1 and BAZ1B depletion will result in UV hypersensitivity. SNF2H recruitment to sites of DNA damage initiates with rapid accumulation at the damage site followed by re-distribution to sites at the periphery of the damage, where it associates with BAZ1B through its SANT-like ISWI (SLID) domain and induces chromatin remodeling effects [38].

Histone 2AX is a variant of histone 2A that comprises almost 10% of the overall H2A in the cell, contains regulatory phosphorylation sites within its unique C-terminal segment, and is phosphorylated at DNA damage sites within seconds after DNA double strand break (DSB) induction in mammalian cells [75]. It is involved in maintaining genome stability as a tumor suppressor gene and is associated with cell death through its major role in the DNA damage response [76]. Ionizing radiation-induced foci (IRIF) are hallmarks of DSB and are enriched for DNA repair and checkpoint proteins as well as γH2AX, which denotes that histone H2AX is phosphorylated at serine 139 (Ser139). IRIF are illuminated by the localized acquisition of γH2AX through the activities of the ATM and ATR kinases [77,78]. H2AX function is actively regulated by reversible phosphorylation not only at the C-terminal Ser139 but also at tyrosine 142 (Tyr142).

### 3.1. BAZ1B Atypical Kinase

Heterochromatin-specific regulation of DNA damage responses involve the enrichment of H2AX and the WICH complex within H3K9me3-containing heterochromatin regions, where BAZ1B can preferentially mobilize H2AX mononucleosomes and phosphorylate Tyr142 (Figure 8). BAZ1B is involved in DNA damage-mediated chromatin remodeling, and its depletion impairs γH2AX and ATM foci formation. Tyr142 is present in metazoans but not in unicellular eukaryotes and is constitutively phosphorylated under basal conditions in cells lacking DSB damage. Tyr142 phosphorylation is not a prerequisite for Ser139 phosphorylation, but they do cooperate to regulate DNA damage repair or to induce apoptosis. It also seems to play a key role in maintaining or promoting pSer139 [27,79,80].

Although BAZ1B catalyzes ATP + L-tyrosyl-[protein] = ADP + H^+^ + O-phospho-L-tyrosyl-[protein] as a phosphorylation reaction in a Mn^2+^ (but not Mg^2+^) dependent manner; there is no shared homology between known kinase domains and BAZ1B. Through bioinformatic and biochemical analysis of recombinant truncated BAZ1B proteins, two putative, well-structured motifs were reported in the N-terminus part of BAZ1B as the kinase domain. These adjacent motifs were named N-motif and C-motif. Their co-expression was found to be required for restoring optimal kinase activity. The N-terminal motif contains the previously described WAC domain, and its expression alone is sufficient to display kinase activity although only at minimal levels. The C-terminal motif, however, does not have kinase activity, but it enhances the activity of the N-motif and thus is required for the optimal kinase activity of the domain [27]. Eyes absent homolog 3 (EYA3) phosphatase shows specificity to H2AX and preferentially dephosphorylates pTyr142.H2AX in a chromatin context while showing no specificity to pSer139.H2AX [75,81].

### 3.2. Phosphorylated Tyr142

The existence of previously hypothesized di-γH2AX (pSer139/pTyr142.H2AX) was first confirmed in 2012 by Singh et al.; it exists in the very early states of the DNA damage response (DDR). The authors showed that mediator of DNA damage checkpoint protein 1 (MDC1) binds canonical γH2AX in DDR pathways but displays no affinity toward di-γH2AX. On the other hand, the microcephalin (MCPH1) protein reads both pSer139.H2AX and pSer139/pTyr142.H2AX modifications via its BRCA1 C terminus (BRCT) domain. MCPH1 is part of a large multiprotein complex and is involved in early DDR. MCPH1 foci formation happens in a γH2AX-dependent manner and has been implicated in chromatin remodeling (through interaction with WICH), signaling, and DNA repair pathways interacting with BRCA2 and condensin II [82]. In normal growth conditions, Tyr142 is phosphorylated by BAZ1B, and upon DNA DSB induction, H2AX is instantaneously phosphorylated on Ser139 and is progressively dephosphorylated on Tyr142; thus, H2AX transitions from being in a diphosphorylated to a monophosphorylated state [82,83].

It has recently been reported that H2AX loss significantly impacts erythroid lineage. H2AX knock-out mice show macrocytic anemia and dysregulation of the gene programs associated with late erythroid maturation. A larger proportion of the population are immature erythroblasts with an increased nuclear:cytoplasmic ratio [84]. These cells accumulate less hemoglobin in the first 4 days of maturation and cannot condense their nuclei compared to WT cells. Caspase-initiated chromatin condensation has been shown to be impaired in these cells, resulting in the dysregulation of terminal erythroid maturation. H2AX shows specific dynamics during erythropoiesis, and its phosphorylation at both Ser139 and Tyr142 is regulated in terminal erythroid maturation. In HUDEP-2 (erythroid model cells), BAZ1B is robustly produced at day 0 of maturation and disappears by day 7. Interestingly, BAZ1B knock-out HUDEP-2 cells start with normal maturation and proliferation phenotypes; however, by day 7 of maturation, large and less condensed nuclei with increased cell size resembling the same terminal maturation defects as H2AX knock-out was observed. pY142-H2AX is lost in these cells, but pS139 is present. Full length caspase-3 persists during maturation and does not undergo complete cleavage in the absence of BAZ1B, resulting in impaired chromatin condensation. Although H2AX-knock-out nuclear condensation defects are present in BAZ1B knock-out cells, hematopoietic and erythroid genes are not affected in contrast to H2AX knock-out. This suggests that BAZ1B and the subsequent Tyr142 phosphorylation on H2AX is required for proper chromatin condensation during erythroid maturation.

### 3.3. BAZ1B and Apoptosis

Additional evidence of pTyr142.BAZ1B involvement in apoptosis induction was reported in K562 human leukemia cells. Imatinib mesylate has been used successfully to treat chronic myelogenous leukemia (CML) and is a selective inhibitor of small-molecule protein kinases, which, in the case of CML, target the BCR/ABL fusion protein kinase. Imatinib also induces the diphosphorylation of H2AX at Ser139 and Tyr142 residues and activates the caspase-3/Mst1 apoptotic pathway. Enhanced BAZ1B protein levels were observed in imatinib-treated cells, and therefore, it is plausible that Tyr142 phosphorylation induced by imatinib is dependent on BAZ1B; however, the mechanism by which BAZ1B levels are increased in imatinib treatment is not yet understood [85].

Furthermore, BAZ1B involvement in a viral infection-related apoptosis was reported in a model. Adenovirus infection is regulated by its E4 open-reading-frame 4 (E4orf4) protein, which can induce a unique non-classical programmed cell death in transformed cells associating with protein phosphatase 2A (PP2A) and Src kinase. PP2A phosphatase and Src kinase are two significant partners of E4orf4 that can reenforce its pro-apoptotic activities. E4orf4 induces PP2A recruitment to chromatin and targets it to the ACF complex (Acf1/hSNF2H). In the model proposed by Brestovitsky in 2011, PP2A recruitment to chromatin results in the dephosphorylation of chromatin substrates and induces chromatin structural changes, which are required for proper chromatin remodeling, followed by altered cellular functions such as DNA transcription and replication. E4orf4 has functional interactions with both members of the ACF complex. SNF2H knock down inhibits cell death; however, Acf1 knock down increases E4orf4-induced cell death. Therefore, SNF2H activity and Acf1 inhibition are required for cell death induction in this model. According to the model, the inhibition of Acf1-containing complexes shifts the balance toward other SNF2H-containing complexes such as WICH and their subsequent overactivity. This part of the model is supported by the determination that shRNA-mediated BAZ1B knock down prevents E4orf4-induced cell death since it prevents the shift from the inhibited ACF complex toward the WICH complex [86]. While the model fails to explain the underlying reason why the shift from ACF toward WICH leads to cell death, it might be due to increased Tyr142 phosphorylation caused by BAZ1B and the subsequent increased tendency toward apoptosis rather than DNA repair.

## 4. BAZ1B and Neural Development

*Xenopus laevis* embryos display defects in the development of several organs such as embryonic anterior neural crest due to BAZ1B knock down. Neural crest migration and further survival are severely impaired since BAZ1B depletion results in increased apoptosis within the hindbrain, more specifically, within the neural crest migration path. These data highlight a central role for BAZ1B in neural development and proper neural crest functioning [87].

Such embryos also show significant over-expression of bone morphogenetic protein 4 (BMP4) [87], which is a highly conserved factor in vertebrates and is involved in epidermis formation, mesoderm patterning, and the suppression of neural differentiation [88]. In addition to BMP4 up-regulation, the down-regulation of a number of regulatory factors has also been reported [87], including the neural cell adhesion molecule (NCAM, CD56), which is expressed on the surface of neurons [89]; sonic hedgehog (SHH), which regulates embryonic development as a signaling molecule [90]; myogenic regulatory factor 4 (MRF4), which is involved in skeletal muscle differentiation [91]; paired box gene 2 (PAX2), which is a transcription factor involved in the embryonic development of organs and systems such as eyes and the central nervous system (CNS) as well as kidney cell differentiation [92]; ephrin type-A receptor 4 (EPHA4), which is crucial for development of dendritic spines [93]; and SRY (sex determining region Y)-box2 (SOX2), which is essential in CNS development and is an invaluable transcription factor orchestrating embryonic stem cell self-renewal and pluripotency [94]. It has been reported that neural differentiation defects due to BAZ1B haploinsufficiency can be rescued by antagonizing the over-active Wnt signaling pathway in neural stem cells. BAZ1B target genes are enriched within the Wnt signaling pathway that regulates the balance between neural progenitor self-renewal and differentiation. Thus, BAZ1B haploinsufficiency shifts away from differentiation toward prolonged self-renewal. Significant transcriptional dysregulations in neural progenitor cells can also result from BAZ1B haploinsufficiency [95]. An interesting question is what the molecular basis for the expression profile disruption is for certain genes that result in an imbalance between self-renewal and differentiation and subsequent developmental defects. The orchestrated antagonistic activities of BMP4 and SHH, which are mis-regulated in BAZ1B knockdown embryos, are also crucial in the normal development and formation of many neural structures such as the eyes and the brain [96].

Homozygous BAZ1B mutations in mice result in skull deformity, with these mice exhibiting a flattened nasal bone, protruding forehead, shorter snout, and upward curvature of the nasal tip, which are similar to the facial features observed in WS patients [97]. Infantile hypercalcemia and ocular defects that are common in WS patients can potentially be explained by BAZ1B deletion since thyroid and ultimobranchial bodies are developed from the branchial apparatus and eye tissues (including cornea, endothelium, stroma, and muscles of the iris) and are generated from neural crest cells [87].

Distinguished craniofacial and prosocial features have been described for anatomically modern humans. These alterations are similar to the traits of domesticated species compared to their wild types and the root human self-domestication hypothesis [98]. A large group of genes, including BAZ1B, has been highlighted in the modern human genome to be enriched for frequent non-synonymous changes compared to archaic ones (Neanderthals and Denisovans). Most of these changes fall in regulatory regions of BAZ1B as well as its downstream targets [99,100]. This is the first outstanding validation for the human self-domestication hypothesis proposing BAZ1B as one of the genes in regulating the modern human face. Recently, BAZ1B contribution to in vitro neural crest stem cell induction and distal regulation have also been reinforced, and gene activity in the WS critical region, including BAZ1B, are highlighted to be involved in domestication events in canids [101].

Transcriptional alterations due to BAZ1B dosage effects are important for the pathways that are critical in neural crest development, maintenance, and downstream skeletal and cardiac outputs. In neural crest stem cells, BAZ1B preferentially binds to distal regulatory regions and is involved in NCSC enhancer regulations [100]. It has been hypothesized that domestication events in the course of evolution are rooted in mild neural crest cell defects, and BAZ1B is proposed among the genes potentially contributing to it [102].

## 5. BAZ1B and Other Outstanding Roles

In addition to the well-established roles of BAZ1B in the three main DNA functions (replication, repair, and transcription) and the growing knowledge of BAZ1B involvement as an effective factor in neural development, other central functions have also been attributed to this versatile protein.

### 5.1. BAZ1B and Chromosomes

The ACF and WICH complexes have been reported as major components of mitotic chromosomes in *Xenopus* eggs [103]. There are several non-histone proteins involved in forming and maintaining structurally stable mitotic chromosomes. These include topoisomerase IIa (TOP2A), topoisomerase I (TOP1), and structural maintenance of chromosomes protein 2 (SMC2) as part of the condensin complex [104]. BAZ1B has been reported to promote TOP1 activity during replication by providing it access to replication forks [105] and is also enriched throughout chromosomes in the early stages of condensation as cells approach mitosis [31]. BAZ1B localizes to the mitotic chromosome axis before mitotic entry, and its absence results in alterations of chromosome condensation timing and subsequent prophase delay and mitosis progression errors; however, these effects are not fatal, and instead, slow cell growth and chromosome condensation will eventually be achieved. These effects are more profound in the case of a BAZ1A/B double deletion, which, similar to the cells lacking condensin I or II, display frequent chromosomal segregation errors. Based on this evidence, BAZ1 proteins have been proposed to expedite the association of other chromosome scaffold proteins via nucleosome spacing and, in this way, employ its roles in maintaining chromosome structure [106].

### 5.2. BAZ1B and Cancer

BAZ1B phosphorylation at serine 158 (Ser158) significantly contributes to breast cancer tumorigenesis. Although the exact identity of the kinase that phosphorylates BAZ1B is unknown, the activation of the Ras/MAPK signaling pathway in MCF-7 cells results in BAZ1B phosphorylation. In MEC-7 nuclei, BAZ1B colocalizes with phosphorylated ERK1/2, which has been suggested as a possible kinase for BAZ1B phosphorylation [107]. The Ras/MAPK signaling pathway is misregulated in a variety of human cancers, where its activation phosphorylates downstream oncoproteins [108]. Precisely if and how BAZ1B phosphorylation contributes to oncogenesis and the possible effects that BAZ1B phosphorylation might have on chromatin remodeling activity and subsequently its impact on replication, repair, and/or transcription have yet to be determined. If BAZ1B phosphorylation leads to its misregulation where its normal activity is required, such as in replication and transcription, then its plausible that its abnormal function is in favor of cancer progression.

Regardless of a possible role for BAZ1B phosphorylation, the acetylation of BAZ1B has been reported to promote the expression of tumor-related genes, increase cell proliferation, and promote tumor formation. BAZ1B lysine 426 acetylation (BAZ1B-K426ac) enhances its kinase activity and results in increased levels of pTyr142-H2AX. This is in contrast to the normal processes in which Tyr142 dephosphorylation should result in the promotion of γH2AX enrichment and the repair response. BAZ1B-K426ac is mediated by the acetyltransferase MOF (males-absent on the first protein) [109], the dysregulation of which has been highlighted in many types of human cancer [110]. BAZ1B-K426ac levels are significantly associated with MOF levels and are high both in breast cancer cell lines and in patient tissues.

The acetylation of BAZ1B positively impacts its phosphorylation at serine 158, enhancing its interaction with H2AX and ultimately resulting in the elevated levels of pTyr142-H2AX that promote cell growth. The overexpression of MOF and/or BAZ1B as well as a mutant the acetylation-mimicking variant of BAZ1B all result in cancer cell proliferation and migration. BAZ1B deacetylation is conducted by SIRT1 (Sirtuin 1) [109], which is a well-established NAD-dependent protein deacetylase, acting on various substrates involved in important cellular functions. An interesting point about SIRT1 is its regulatory involvement in biological processes that are acting antagonistically. For instance, it acts as a hub and interacts with complex network of proteins that are involved in both cancer development and autophagy events [111].

Further evidence of an oncogenic role for BAZ1B was reported in normal human intestinal epithelial cells that were overexpressing an exogenous copy of an oncogenic form of KRAS where glycine 12 was altered to a valine (KRASG12V). These cells showed activation of the MAP kinase pathway and inappropriate expression of neuregulin-3 (NRG3). Neuregulins are members of the superfamily of epidermal growth factor (EGF) ligands and are markedly expressed in solid tumors and bind to the ErbB/HER receptor tyrosine kinases, activating intracellular signaling pathways [112]. It was found that BAZ1B was present in the cell media, and that the overexpression of NRG3 alone was sufficient to cause this. Here, inappropriately activated NRG3 was found to interact with a region in the amino-terminal portion of BAZ1B (between amino acids 1–576), and media containing NRG3-BAZ1B were capable of eliciting signaling pathways and irreversibly activating endogenous NRG3 expression in normal cells. Interestingly, NRG3-BAZ1B was detected in urine and serum from a large cohort of colon cancer patients who carried KRASG12 mutations. Knocking down BAZ1B or treating cells with antibodies to BAZ1B reduced cell proliferation and oncogenic pathway activation, implicating BAZ1B as a target for KRASG12 colon cancer [113].

Furthermore, BAZ1B was reported in lung cancer as an oncoprotein, where its overexpression promotes proliferation, colony formation, migration, and invasion of lung cancer cells as well as tumor growth and invasion in mouse xenograft models [114]. In addition to the activation of both of the PI3K/Akt and IL-6/STAT3 oncogenic pathways, the expression of several cancer-related genes including Fibronectin (FBN1), Fos (Fos proto-oncogene, AP-1 transcription factor), and CEACAM6 (carcinoembryonic antigen-related cell adhesion 6) are upregulated as a result of BAZ1B overexpression. These genes are involved in the induction of the epithelial to mesenchymal transition (EMT). Thus, BAZ1B likely promotes metastasis in lung cancer via the induction of EMT (Meng et al., 2016). BAZ1B acetylation promotes the overexpression of EMT-inducing genes (FBN1, Fos, and CEACAM6) at mRNA levels, whereas these mRNA levels are decreased in the case of deacetylated BAZ1B; therefore, MOF acetyltransferase might induce its oncogenic effects by acetylating BAZ1B as one of its downstream targets [109].

### 5.3. BAZ1B and Behavior

Chromatin remodeling complexes have been reported to be involved in the formation of complex brain behaviors. In two cases of chronic emotional stimuli in mice, including chronic cocaine treatment and chronic social defeat stress, BAZ1B is upregulated in the nucleus accumbens (NAc), which is a critical brain reward region. The overexpression of BAZ1B and Snf2h in mouse NAc potentiates the response to cocaine self-administration and tolerance to chronic social defeat stress. Increased BAZ1B and Snf2h in NAc promotes behavioral responses to cocaine in noncontingent and contingent situations. It can elevate rewarding stimuli responses in the brain; however, BAZ1B upregulation is transient and does not persist during behavioral effects and returns to basal levels within hours or days, indicating the possibility that BAZ1B regulates the downstream genes involved in persistent behavioral effects. These findings suggest empirical roles for BAZ1B as well as its associated chromatin remodeling complex, WICH, for regulating reward behaviors and their stimulus-specific nature of action [114].

### 5.4. BAZ1B and Noise-Induced Damage

In an attempt to investigate the early proteomic changes involved in hearing loss following the exposure to deteriorating noise, Jamesdaniel et al. highlighted the initiation of cell death in sensory epithelium and modiolus following noise induction. The P38/MAPK signaling pathway has been proposed to transmit noise-induced stress signals. Interestingly, BAZ1B upregulation was reported along with six other proteins in sensory epithelium as well as in cochleae that trigger the noise-induced cell death. Anatomical studies have provided supporting evidence for the possible interaction between BAZ1B, p38/MAPK, E2F3, and phosphorylated FAK, all of which were expressed in outer hair cells.

Noise-induced damage can be limited by the activation of the BAZ1B-mediated DNA repair mechanism. However, based on immunoblotting and microarray assays, it has been speculated that BAZ1B may have additional non-nuclear functions in response to noise-induced damage [115]. Although further studies are required to elucidate BAZ1B involvement in noise-induced hearing loss, taking its major chromatin regulatory activities into account, BAZ1B seems to function within the nucleus by mediating downstream gene expression changes that conduct noise-mediated damage signals to the cells.

## 6. Conclusions and Further Insights

The very basic definition of “chromatin remodeling” indicates the action of chromatin remodeling enzymes on moving, ejecting, destabilizing, and restructuring nucleosomes on chromatin fibers [116,117]. It involves dynamic interchanges between open and closed chromatin structures. Such chromatin restructuring can locally regulate transcriptional processes, considering that open chromatin is more commonly associated with a transcriptionally active state and that closed chromatin is typically considered a more transcriptionally repressed state. Spreading along fibers, remodeling processes can also modulate fiber condensation through long-range effects. Chromatin remodeling is regulated at three levels: (1) sequence preference of the histone octamer [118], (2) nucleosome identification by chromatin remodeling complexes, and (3) the final motor action of the remodeler on the nucleosome. Unlike many other molecular motors that simply transport cargo, chromatin remodelers do not move nucleosomes in a linear fashion [119]. Obviously, unwrapping nucleosomal DNA is a more complex action than simple linear transport.

BAZ1B mediates many key chromatin remodeling events as an accessory subunit component of two well-established chromatin remodeling complexes that orchestrate various molecular functions including but not limited to nascent chromatin formation following appropriate DNA replication, facilitating RNA polymerase (I, II and III) activity by providing open chromatin and DNA accessibility, regulating the DNA damage repair response by inducing a specific protein modification on H2AX nucleosomes, and helping the cell to make the decision between life or death. Figure 9 summarizes BAZ1B involvement in the many important cellular functions and other accompanying factors related to BAZ1B functioning (Figure 9). Furthermore, empirical evidence highlighting BAZ1B function in proper neural development conveys probable genotype–phenotype correlations between BAZ1B copy number variation and neurodevelopmental defects found in Williams syndrome patients (single copy) and 7q11.23 duplication syndrome patients (three copies). Recent findings also attribute BAZ1B as playing a central role in maintaining the mitotic chromosome structure as well as involvement in complex traits such as cancer progression and even animal behavior. BAZ1B indeed deserves to be fully understood since considering its profound and decisive roles, it is very probable that this versatile protein is involved in other critical cellular functions that have yet to be discovered.

## Figures and Tables

**Figure 1 genes-12-01541-f001:**
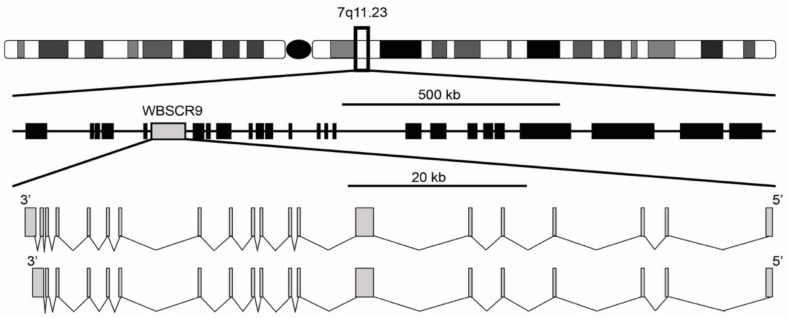
Genomic organization of WS deletion region including WBSCR9 gene. There are two isoforms of BAZ1B gene on DNA minus strand with either 19 or 20 exons.

**Figure 2 genes-12-01541-f002:**
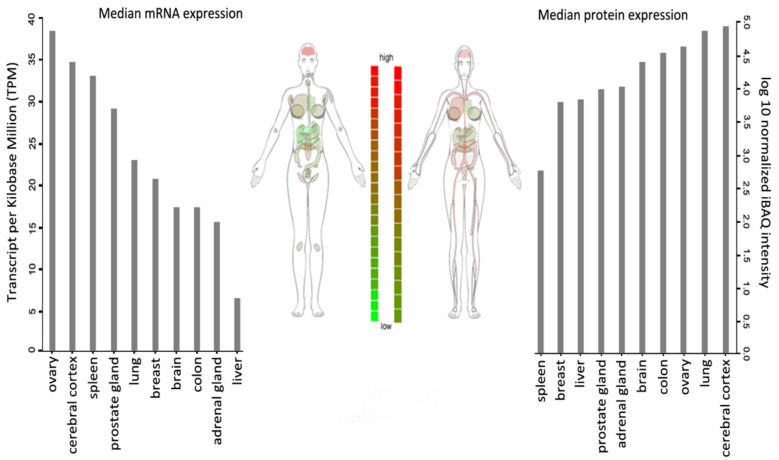
BAZ1B expression in different tissues. Images and expression data were obtained from proteomics database [8,9]. Body visualization is a female body. Proteomics data comes from mass spectrometry (MS1) analysis. Transcriptomics data comes from RNASeq analysis.

**Figure 3 genes-12-01541-f003:**
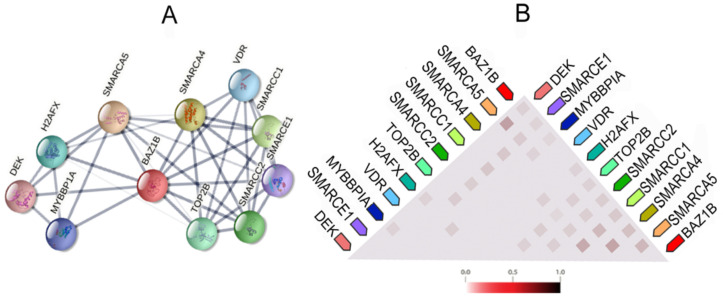
BAZ1B co-expression and interactions. Figures are obtained from STRING database [10]. (**A**) Each node in the network represents all proteins produced by a single, protein-coding gene locus, i.e., splice isoforms or posttranslational modifications are collapsed. Edges represent protein–protein association, which are meant to be specific and meaningful. Proteins jointly contribute to a shared function; however, this does not necessarily mean they are physically binding each other. Line thickness indicates the strength of data support. (**B**) Co-expression of BAZ1B and first shell of interactors. The color intensity in the triangular matrices indicates the level of confidence that two proteins are functionally associated, given the overall expression data in the organism.

**Figure 5 genes-12-01541-f005:**
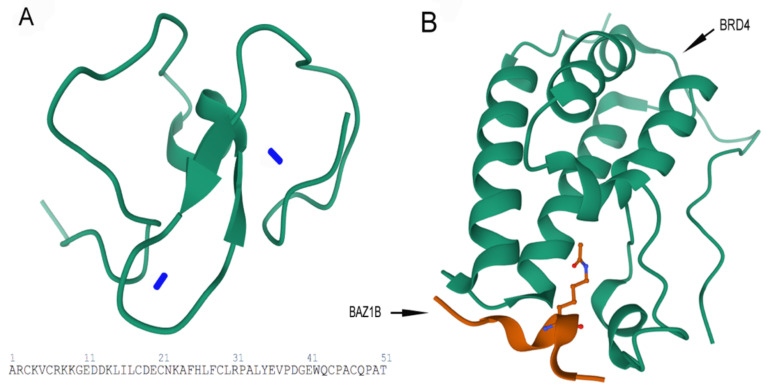
Images are obtained from the protein data bank. The structure information comes from primary publication [26]. (**A**) Tertiary structure of 51 residues in PHD zinc finger domain of BAZ1B with the two zinc ions (Zn^2+^) (blue sticks) are shown (residues 1185–1235). PDB ID: 1F62. (**B**) Crystal structure of first bromodomain of human BRD4 in a complex with an acetylated BAZ1B peptide (residues 217–226; FLPH(ALY)YDVKL; K221 ac). PDB ID: 5NNF.

**Figure 6 genes-12-01541-f006:**
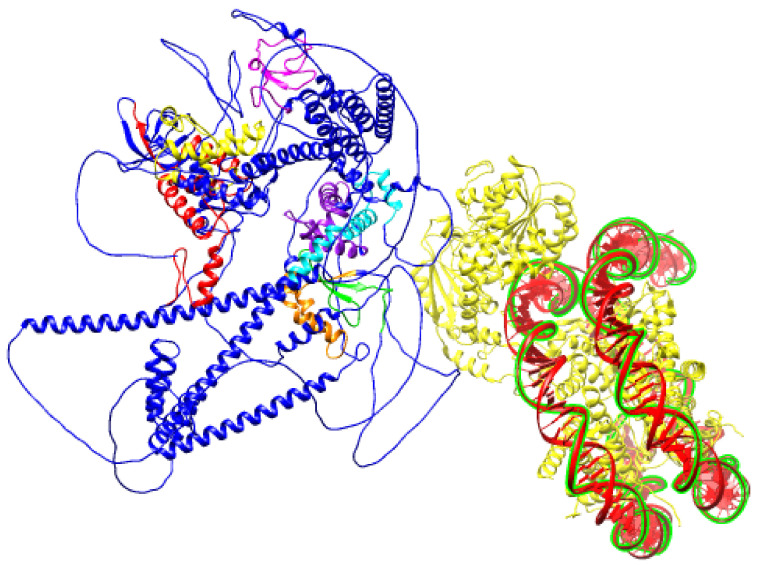
Schematic prediction of SNF2H SANT domain bound to DNA (PDB ID:6NE3) and BAZ1B protein (PDB ID: Q9UIG0) facing it with an ISWI binding pocket comprised of DDT, WHIM1, WHIM2, and WHIM3 domains.

**Figure 7 genes-12-01541-f007:**
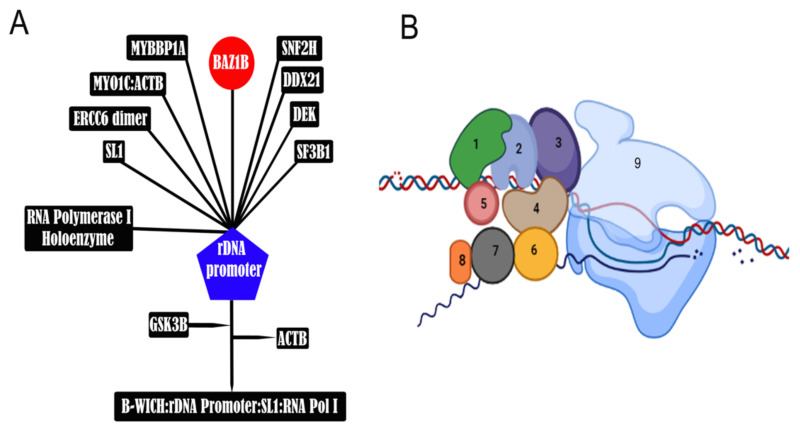
B-WICH is involved in H3K9 acetylation. (**A**) B-WICH multiprotein complex along with SL1 transcription factor binds active rRNA gene promoters forming the B-WICH:rDNA Promoter:SL1:RNA Pol I complex. Upon the formation of this complex, the ERCC6 (CSB) and MYO1C from the B-WICH complex interact with histone acetyltransferases, including KAT2B (PCAF), which acetylates H3K9 as well as KAT2A (GCN5) and KAT3B (EP300). MYO1C binding to chromatin is dependent on phosphorylation by GSK3beta; Note that MYO1C is initially bound to *β*-Actin (ACTB), which is released after GSK3B phosphorylates MYO1C. MYO1C binding induces open chromatin formation in a 200bp region on rDNA promoter [70]. Activation and maintenance of RNA Pol I transcription requires MYO1C recruitment to active rRNA genes followed by B-WICH binding that will recruit histone acetyltransferases. (**B**) Schematic representation of B-WICH complex next to RNA polymerase in active transcription phase. 1: BAZ1B; 2: SN2H; 3: MYO1c; 4: CSB; 5: DEK; 6: MYBBP1a; 7: SF3b1; 8: DDX21; 9: RPN. Note that the interaction of DDx21, MYBBP1a, and SF3b1 with B-WICH complex is RNA-dependent.

**Figure 8 genes-12-01541-f008:**
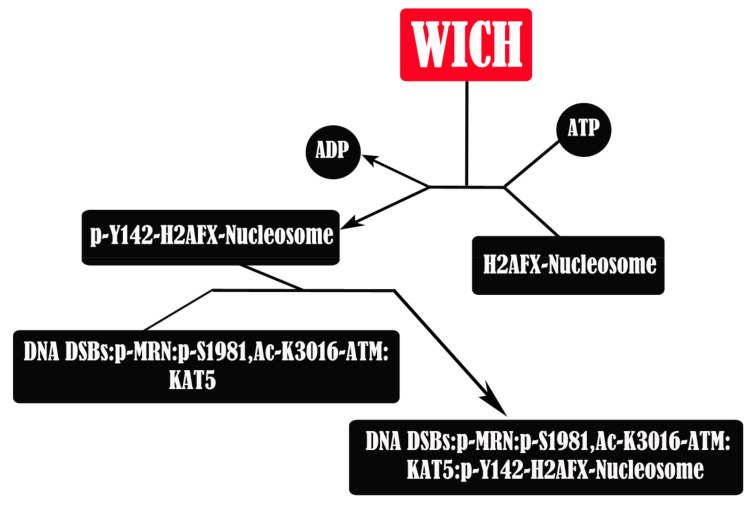
BAZ1B is involved in the recruitment and ATM-mediated phosphorylation of repair and signaling proteins at DNA DSBs. It phosphorylates H2AF on Tyr142. After DNA DSB induction, the MRN complex (Mre11, Rad50, and Nbs1) binds to the DNA breaks followed by ATM recruitment. ATM, in dimeric form, is in a complex with KAT5. KAT5 then acetylates ATM at DNA DSB. MRN activates ATM dimer by promoting its dissociation into monomers and then ATM phosphorylates the NBN (Nbs1) component of the MRN complex, resulting in the p-MRN:p-S1981,Ac-K3016-ATM:KAT5 protein complex at DNA DSB sites. This complex then recognizes and associates with the p-Y142-H2AFX-nucleosome. Phosphorylation of Ser139 and dephosphorylation of Tyr142 are the following steps. This nucleosomal complex will then undergo multiple modifications such as phosphorylation, methylation, and ubiquitination as well as binding of additional proteins and will eventually lead into nonhomologous end-joining (NHEJ) and homology-dependent repair through homologous recombination repair (HRR) or single strand annealing (SSA).

**Figure 9 genes-12-01541-f009:**
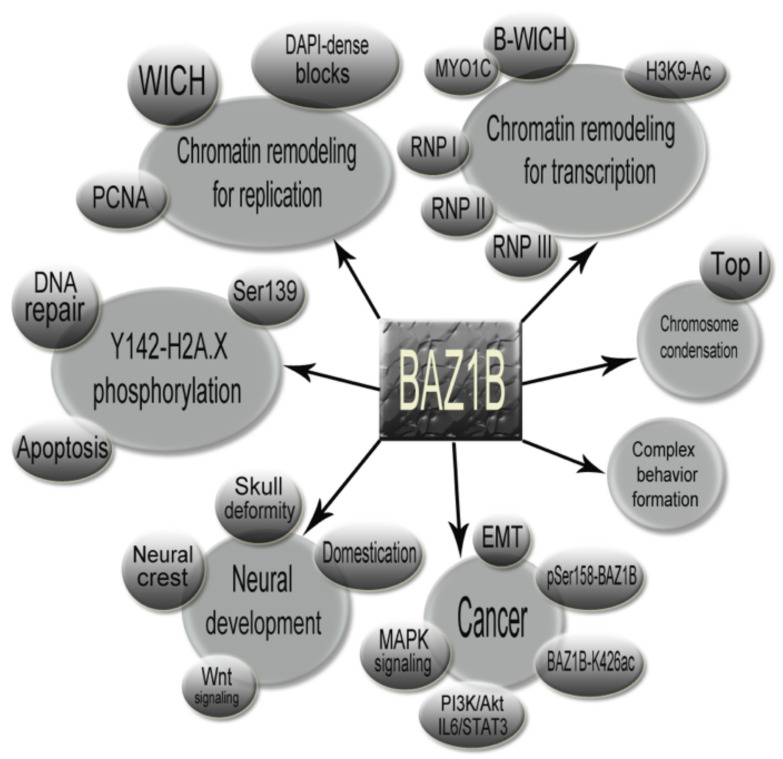
Concept map illustrating the major functional events mediated by BAZ1B as well as other related events and factors that are important to be considered.

## Data Availability

Not Applicable.

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
