# Peer review of "BAZ1B the Protean Protein"

_genes, 2021, doi:10.3390/genes12101541_

Round 1

Reviewer 1 Report

The review article titled “BAZ1B the protean protein” provides detailed information on the structure and function of BAZ1B and an overview of the physiological and molecular consequences of known deletions.  The review is organized and comprehensive. A suggested addition to the review is the role of BAZ1b in erythroid maturation and chromatin condensation during terminal through H2A.X Y142 phosphorylation (Nazish, et al. 2021). The authors do mention the kinase activity of BAZ1B, but the specific consequences of H2A.X Y142 phosphorylation seem to be missing.  However, the review provides adequate information on the effects BAZ1b on chromatin condensation aside from this particular mechanism.

There are a few formatting issues to address such as font size changes that occur in lines 55/56 among others. 

Nazish, JN. Et al. 2021. Histone H2A.X phosphorylation and Caspase-Initiated Chromatin Condensation in late-stage erythropoiesis. Epigenetics Chromatin. 2021; 14: 37.. https://dx.doi.org/10.1186%2Fs13072-021-00408-5

Author Response

  • We thank the reviewer for their comments and are grateful for their careful eye picking up upon the font issue. The font has been corrected and the recommended manuscript (New citation 84) has been incorporated into the revised manuscript (Lines 584-603).

Reviewer 2 Report

The paper is well-written. The review is interesting and deeply describe Bromodomain Adjacent to Zinc finger domain 1B. I have no recommendations or questions for the Authors.

However, the Authors should consider adding some phrases that suggest the article's review character in the title.

Please, verify the keywords interpunction also. 

Author Response

  • We are grateful to the reviewer for their kind words and thank them for picking up upon the interpunction error in the keywords section. This has been corrected. While we did think hard about how best to adjust the title, after much discussion we respectfully kept the title as is, as we were unable to come up with an alternative that was not superfluous to the existing title.

Reviewer 3 Report

This review manuscript is decently written as per the text goes. However, it requires major changes pertaining to the organization of the text and it also requires several key figures.

MAJOR COMMENTS

  1. There are many instances where the focus of the review is significantly deterred from BAZ1B by long and convoluted details of the field. While these seem quite appropriate for the discussion in #5 and #6, there is barely any need for similar expanded discussion in the beginning of each section (especially in #2 and #3 as well as their subsections). If the authors can refurbish this section, like #4 (well written), it would make the document way stronger. In general, it would also be prudent to condense the introductory background details for each section and move some of them to the relevant discussions of #5 and #6.

  1. Figures from this entire document needs major upliftment:
  • Fig 4: requires upgrade on colors, maybe larger boxes to highlight the domains. Also, individually there should be representative inset images of the structures of these individual domains.
  • The bromodomain section requires a structure highlighting the interaction between the Asn and Acetyl groups.
  • Fig 5: requires Zn+2 interactions shown as sticks and PDB IDs should be added in general wherever structures are used.
  • Need a schematic representation of DDT and three WHIM motifs in interaction between Acf1 and SNF2. Otherwise, the entire paragraph seems utterly confusing. #1.5
  • In #2.2 a B-WICH complex schematic with possible highlight of BAZ1B will be crucial, especially if it can be adapted as a second panel to Fig 6. The interactors that are RNase resistant should also be highlighted to complement the text.

  1. The review also requires an all-encompassing figure to highlight all the different functions associated with BAZ1B protein. It can be included around the last two sections, to complement the functional discussion.

MINOR COMMENTS

  • Line 574: reinforce – Spelling error.
  • Line 122: What is ‘25’ -- unclear
  • Line 121: The figures need actual labels of the protein in the figure.

Author Response

  • We thank the reviewer for their detailed and thoughtful critique and provide point-by point responses below each comment.

MAJOR COMMENTS

  1. There are many instances where the focus of the review is significantly deterred from BAZ1B by long and convoluted details of the field. While these seem quite appropriate for the discussion in #5 and #6, there is barely any need for similar expanded discussion in the beginning of each section (especially in #2 and #3 as well as their subsections). If the authors can refurbish this section, like #4 (well written), it would make the document way stronger. In general, it would also be prudent to condense the introductory background details for each section and move some of them to the relevant discussions of #5 and #6.

  • As suggested, we have deleted extraneous details and moved sections as appropriate. The corresponding edits are indicated in the right margin of the track changes.

  1. Figures from this entire document needs major upliftment:
  • Fig 4: requires upgrade on colors, maybe larger boxes to highlight the domains. Also, individually there should be representative inset images of the structures of these individual domains.
  • The bromodomain section requires a structure highlighting the interaction between the Asn and Acetyl groups.
  • Fig 5: requires Zn+2 interactions shown as sticks and PDB IDs should be added in general wherever structures are used.
  • Need a schematic representation of DDT and three WHIM motifs in interaction between Acf1 and SNF2. Otherwise, the entire paragraph seems utterly confusing. #1.5
  • In #2.2 a B-WICH complex schematic with possible highlight of BAZ1B will be crucial, especially if it can be adapted as a second panel to Fig 6. The interactors that are RNase resistant should also be highlighted to complement the text.

  • We thank the reviewer for these suggestions. As requested, Fig.4 was completely redrawn, and the full structure shown. As we are not structural biologists ourselves, we recruited a structural biologist and due to their contribution to the revise figures, they have been added as a new author. As requested, the Zn interactions have been altered to sticks in Fig.5, and a new Fig.6 has been added with schematic structure of the DDT and WHIM domains and PDB identities included in the legend. Finally, we included a B-WICH schematic in the revised Fig.7.

  1. The review also requires an all-encompassing figure to highlight all the different functions associated with BAZ1B protein. It can be included around the last two sections, to complement the functional discussion.

  • This is a great idea and we have now included a new Fig.9 reflecting this in the manuscript.

MINOR COMMENTS

  • Line 574: reinforce – Spelling error.
  • Line 122: What is ‘25’ -- unclear
  • Line 121: The figures need actual labels of the protein in the figure.

  • Thank you for catching these oversights, we have now addressed these in the revised manuscript.